# How Well can Spaceborne Digital Elevation Models Represent a Man-Made Structure: A Runway Case Study

**Kazimierz Becek** [1,*]👤, **Volkan Akgül** [2], **Samed Inyurt** [2], **Çetin Mekik** [3] **and Patrycja Pochwatka** [4]👤

[1] Faculty of Geoengineering, Mining and Geology, Wroclaw University of Science and Technology, 50-421 Wroclaw, Poland

[2] Department of Geomatics Engineering, Zonguldak Bulent Ecevit University, Zonguldak 67100, Turkey

[3] Department of Geomatics Engineering, Hacettepe University, Ankara 06800, Turkey

[4] Department Environmental Engineering and Geodesy, University of Life Sciences in Lublin, 20-033 Lublin, Poland

\* Correspondence: kazimierz.becek@pwr.edu.pl

**Abstract:** In this case study, an active runway of a civilian airport in Zonguldak, Turkey was used to assess the suitability of spaceborne digital elevation models (DEMs) to model an anthropogenic structure. The tested DEMs include the Advanced Spaceborne Thermal Emission and Reflection Radiometer (ASTER), the Advanced Land Observing Satellite (ALOS) World 3D 30 m (AW3D30), the Shuttle Radar Topography Mission (SRTM)-1″, the SRTM-3″, the SRTM-X, the TanDEM-3″, and the WorldDEM$^{TM}$. A photogrammetric high accuracy DEM was also available for the tests. As a reference dataset, a line-leveling survey of the runway using a Leica Sprinter 150/150M instrument was performed. The selection of a runway as a testbed for this type of investigation is justified by its unique characteristics, including its flat surface, homogenous surface material, and availability for a ground survey. These characteristics are significant because DEMs over similar structures are free from environment- and target-induced error sources. For our test area, the most accurate DEM was the WorldDEM$^{TM}$ followed by the SRTM-3″ and TanDEM-3″, with vertical errors (LE90) equal to 1.291 m, 1.542 m, and 1.56 m, respectively. This investigation uses a method, known as the runway method, for identifying the vertical errors in DEMs.

**Keywords:** Spaceborbe DEM; SRTM; TanDEM; AW3D30; runway method; Zonguldak; suitability assessment

## 1. Introduction

Significant progress in the construction of sensors, methods, and platforms for the measurement of distance, angle, position, navigation, and timing has been observed for the last few decades. A consequence of this development is the shift from in situ surveying to more convenient and cheaper solutions, whereby geodata are acquired from platforms, such as unmanned aerial vehicles (UAV), aircrafts, and satellites, located at a certain distance from the operator. Global navigation satellite systems (GNSS), light detection and ranging (LiDAR), and synthetic aperture radar interferometry (InSAR) are prominent examples of technologies that benefit land surveying and allow for survey-grade measurements of objects' locations and dimensions. Over the last approximately twenty-five years, several attempts have been made to develop a global digital elevation model (DEM) of the Earth's landmasses. There are a few uses for such a consistent, accurate, and global DEM, including applications in the military and in many branches of science. The first such global DEM product was the Shuttle Radar Topography Mission (SRTM) [1]. Since 2004, the SRTM became partially available to the public.

InSAR technology was used to produce the SRTM product. Another model, known as the Advanced Spaceborne Thermal Emission and Reflection Radiometer (ASTER version 3), is also a space-based DEM product but, in this case, photogrammetry was used [2–5] to develop the ASTER product. In addition, there are three recent DEM products available: TanDEM-3″ [6], the WorldDEM[TM] [7], and the AW3D30 [8–10]. The former was developed from the TerraSAR-X/TerraDEM-X satellite data (InSAR method), while the latter was developed from the ALOS data, using photogrammetry. The spatial resolution of these DEMs varies between 12 m and 90 m (at the equator). It is anticipated that the spatial resolution of future DEM products will be even higher, and the vertical and horizontal accuracy will increase. This process will benefit a number of surveying projects that could be performed using remotely-operated data acquisition platforms.

The aim of this case study is to contribute to the body of knowledge on the suitability and fidelity of an engineering structure representation using currently available spaceborne DEMs. While the topic of DEM vertical/horizontal accuracy assessment has already been investigated in several papers, this paper's approach differs from previous studies by the type of reference data used. Reference data sets typically used in investigations of DEM accuracy include a higher accuracy DEM [10], reference benchmarks [3,6], and profiles or cross-sections along a known higher accuracy DEM [11]. However, conclusions drawn from these investigations may be biased due to: a) not considering terrain steepness, b) comparing point elevation against pixel elevation (area), and c) arbitrarily selecting cross-sections without considering the terrain's anisotropy. In the present study, an airport runway was used as a testbed. This engineering structure possesses unique features that eliminate the abovementioned limitations of other types of reference datasets. The preferred features of runways as testbeds for DEM vertical accuracy assessment studies include the horizontal orientation (no slope), dimensions (typically a length >1000 m and a width >15 m), typically homogenous material, and surface roughness. Additionally, relevant for this type of study, the data are in the public domain (e.g., [12]).

The objectives of the present study leading to the aim formulated above include:

- Calculating the statistical indicators of the differences between the surveyed runway surface and the corresponding surface extracted from the investigated DEMs.
- Analyzing the statistics of the differences, and drawing conclusions and recommendations for engineering structure surveyors and operators on the suitability of the freely available or reasonably priced spaceborne DEMs for monitoring anthropogenic or natural features.

In this context, the present study investigates the following DEMs: the ASTER v.2, the 3D30, the airborne photogrammetric digital terrain model (DTM), the SRTM (both 1″ and 3″), the SRTM-X, the TanDEM, and the WorldDEM[TM].

The adopted study method, the runway method (RWYM) [13], possesses the ultimate feature of allowing comparisons of the instrument- or method-induced component of DEM errors, which is a fundamental question to be answered before a DEM source is selected for a given project. The findings of this study indicate that, in a vertical accuracy-wise sense, photogrammetry produces the most accurate results. However, there are some situations where the InSAR method is more suitable.

## 2. Materials and Methods

### 2.1. Area of Interest

The testbed used in this study was a runway (18–36) at Zonguldak Airport, located 8 km to the north of Çaycuma, Turkey (Lat = 41°30′25″ N; Lon = 32°05′23″ E), approximately 12.5 m above the mean sea level (a.m.s.l.). According to the official airport's chart, the geoid undulation is 34.14 m (Earth Gravitational Model 2008—EGM2008 33.489 m). Figure 1 shows the location of the runway.

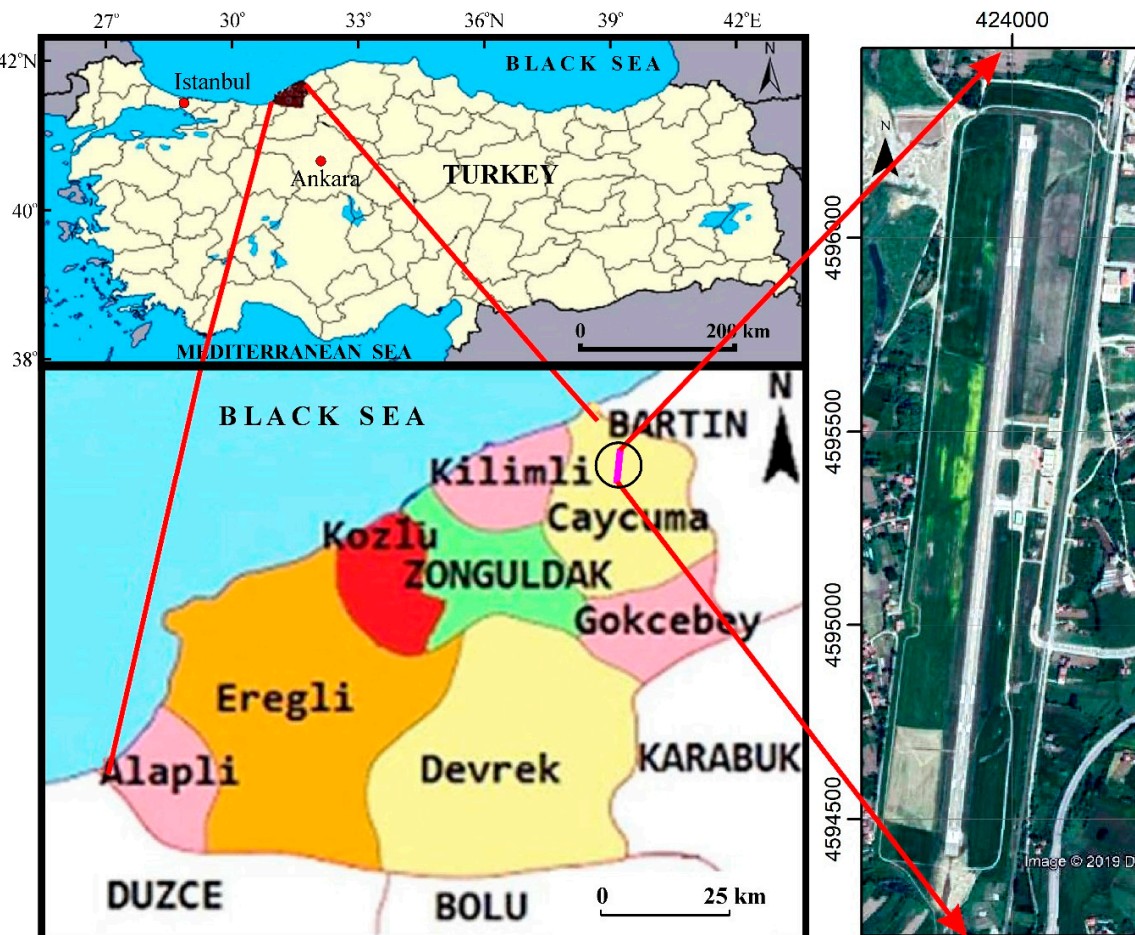

**Figure 1.** Location of Zonguldak Airport, including its runway (18–36). Source: Image: Google Earth®; Maps: own work. Coordinates: WGS84/UTM36T.

Some of the runway's relevant physical parameters are listed in Table 1.

**Table 1.** Selected physical characteristics of the runway used as a testbed in this study. Source: Aeronautical Information Publication (AIP) Turkey.

| Parameter | Value |
|---|---|
| Length (m) | 1881 |
| Width (m) | 30 |
| Threshold elevation a.m.s.l. (18/36) (m) | 12.45/13.3 |
| Surface | Concrete |
| Slope (centerline) | 0.05% |
| Cross-slope (left/right from centerline) | 1.0% |
| Local geoid undulation (m) | 34.14 |

The runway was constructed in the 1960s. Very low air traffic at the airport allowed for easy access to perform the optical leveling. Figure 2 shows the runway's centerline profile and the profile as per aeronautical documentation. It is assumed that the declared profile is an "as constructed" profile of the runway. The maximum discrepancy between the declared and leveled profiles is 0.993 m. While the negative discrepancy between the profiles shown in Figure 2 can easy be explained by land subsidence, the positive discrepancies in the profile's right-hand side is probably due to the recent extension of the runway's length.

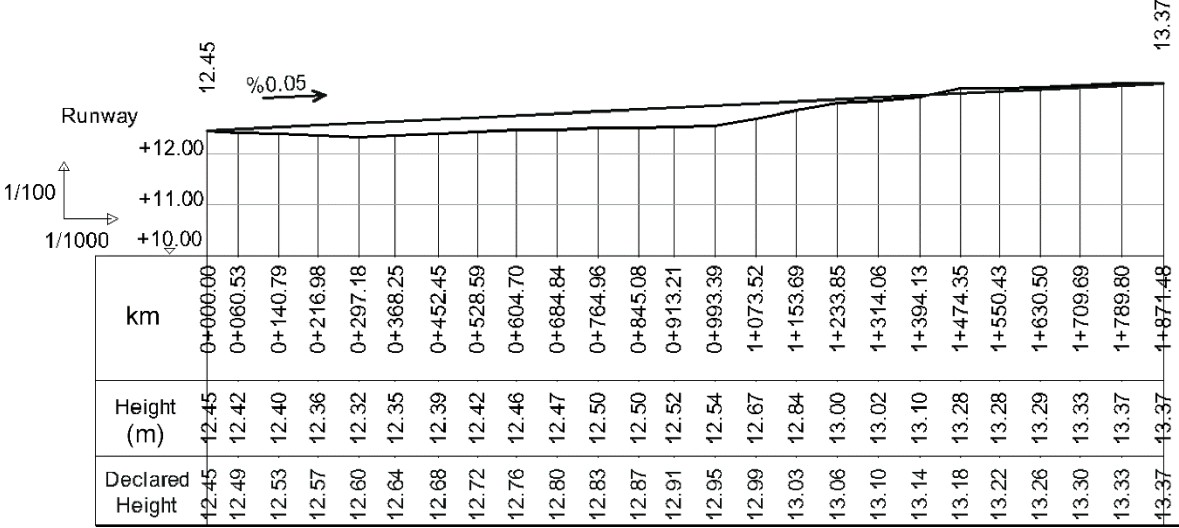

**Figure 2.** Leveled vs. declared centerline runway profile.

Figure 3 shows the leveling of the runway in progress.

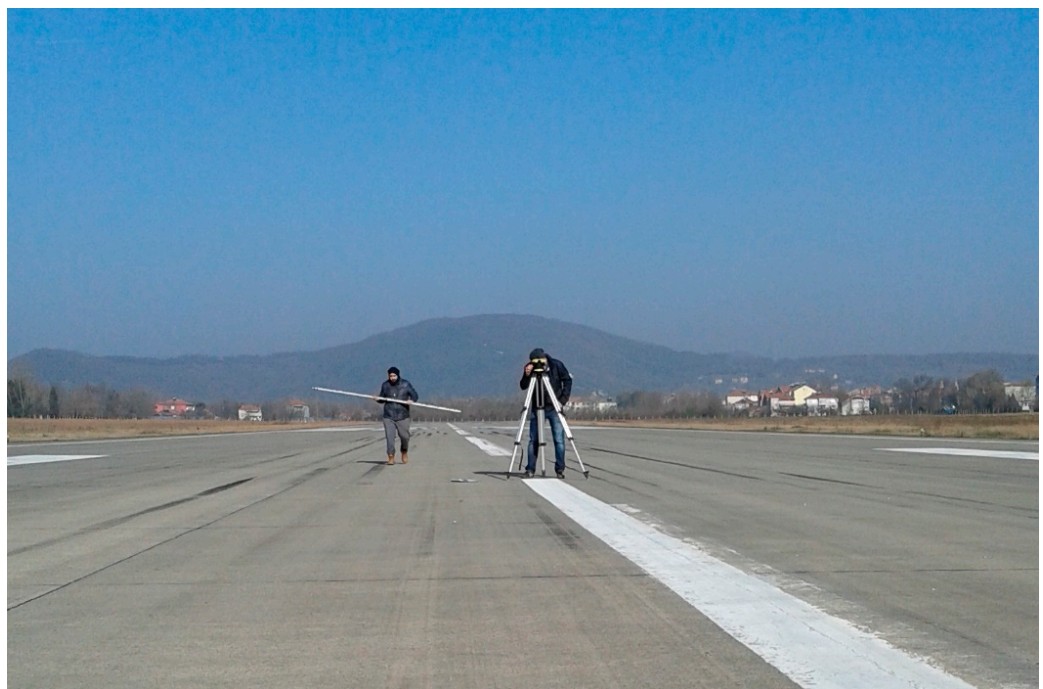

**Figure 3.** Leveling of the runway in progress. Photo: K. Becek.

## 2.2. Digital Elevation Data

In the following sections, the DEMs investigated in this experiment are briefly outlined. The dataset used in this study is available as Supplementary Material from [14].

A common peace of information on geospatial data is the horizontal/vertical reference system used. Table 2 shows reference systems of the DEMs described in the following sections.

**Table 2.** Reference systems used in the investigated digital elevation models (DEMs).

| DEM Brand | Horizontal Reference System | Vertical Reference System |
|---|---|---|
| ASTER | WGS84 | EGM96 |
| AW3D30 | GRS80 (ITRF97) | EGM96 |
| AP | WGS94/UTM36T | Local geoid |
| SRTM-1", SRTM-3" | WGS84 | EGM96 |
| SRTM-X | WGS84 | WGS84 [1] |
| TanDEM, WorldDEM$^{TM}$ | WGS84-G1150 | EGM2008 |

[1] Converted to mean sea level using the local geoid undulation (N = 34.14 m).

### 2.2.1. ASTER

The Advanced Spaceborne Thermal Emission and Reflection Radiometer (ASTER) version 3 is a global DEM. The spatial resolution of the product is 1" (approx. 30 m at the equator). The DEM covers the Earth's surface from 82 N to 82 S. The ASTER version 3 was produced from previous versions by removing outliers, thanks to the inclusion of additional data. The ASTER was produced using the photogrammetry method [15] and processing over 2 million stereopairs. The ASTER data are distributed free of charge in a form of one-by-one-degree tiles [2–5].

### 2.2.2. AW3D30

The AW3D30 is a DEM developed using data captured by the Panchromatic Remote Sensing Instrument for Stereo Mapping (PRISM) installed on board the ALOS satellite. An automatic stereo matching method was used. The AW3D30 data are available for free at a 1 arcsec resolution (approx. 30 m at the equator). The DEM is a resampled version of the original DEM available for free at a 5 m resolution. The AW3D 30 m is distributed as "average" and "median," with the difference in the resampling kernel used. The average version of the AW3D30 is used in the study. The elevations are rounded off to the closest meter and are referenced to the EGM96 geoid. The GRS80 datum is used for the geographic coordinates [7,8].

The target vertical accuracy of the AW3D30 is 5 m (one sigma). This figure seems to be rather pessimistic provided that some researchers demonstrated that the vertical accuracy is 4.10 m (RMSE, root mean square error) [7] or 4.29 m (built-up areas) [11].

### 2.2.3. Aerial Photogrammetry (AP)

For the area of interest (AOI), an aerial photogrammetry-developed DEM was available. The AP model was captured and processed in 2010. The UltraCamX camera was used to capture the aerial photos from an altitude of 1530 m above the ground. The model's pixel size was 10 cm. The vertical accuracy of the model was approximately 10 to 15 cm (one sigma). The project was funded by the Natural Disaster Insurance Institution (TCIP) and carried out by the Directorate General of Geographic Information Systems, Turkey.

### 2.2.4. Leveling

Line leveling was performed using the automatic Leica Sprinter 150/150M Electronic Level Package along with two barcoded leveling staff with a built-in dumpy level and without support. The manufacturer's height accuracy specification is (one standard deviation) 1.5 mm/km double run (ISO 17-123-2). The line leveling was conducted along the centerline of the runway and included intermediate sights on both sides of the centerline (on the edge and in the middle between the edge and centerline). The 25 m distance between the instrument and staff was maintained. The leveling was referenced to a nearby benchmark of Turkey's national heighting system. The leveling loop's disclosure was 3 mm. The leveled spot elevations were used to interpolate 475 spot elevations that

formed a rectangular grid, 9 m × 20 m, oriented parallel to the runway's centerline. These interpolated points were used as control spot elevations in this study.

### 2.2.5. SRTM-1″ and SRTM-3″

The SRTM-1″ and SRTM-3″ elevation data products are perhaps the best known in many branches of science. The data for the DEM were acquired during an 11-day mission of the space shuttle Endeavor in 2000. A technique known as Synthetic Aperture Radar Interferometry (InSAR) was used to develop both DEMs. The products are distributed free of charge at 1″ and 3″ (30 m or 90 m at the equator). The DEM's three-arc second version is a resampled version of the one arc-second version. Many tests of the DEMs [2,16] showed that the vertical accuracy of the SRTM data product is at a level of approximately 2 m (one sigma) or 3.3 m (Linear Error—LE-90%), which is well over the required 16 m. The SRTM-3″ version exhibits less high-frequency noise in elevation because the noise was significantly reduced by the averaging filter applied during the resampling operation. This SRTM version was produced using the Synthetic Aperture Radar (SAR) system working in band C ($\lambda$ = approx. 5.6 cm) of the electromagnetic waves. Both versions of the SRTM DEM were investigated in this contribution.

### 2.2.6. SRTM-X

The SRTM-X is a product of a different instrument than that used for the production of the SRTM DEM flown on board the space shuttle Endeavor during its mission in 2000. The SAR system used the X band ($\lambda$ = approx. 3.1 cm) of the electromagnetic spectrum. The X-band system was jointly developed by the Italian and German space agencies. The SRTM-X product is available at the one arc-second resolution. The SRTM-X elevations were converted to orthometric elevations by subtracting the geoid undulation provided in the aeronautical information on the runway.

### 2.2.7. TanDEM-3″ (TanDEM)

The TanDEM-3″ is a just-published (11/05/2019) DEM data product developed from the TanDEM-X DSM by down sampling to a resolution of 3″ (90 m at the equator). The TanDEM-3″ is a digital surface product available free of charge from https://geoservice.dlr.de/web/dataguide/tdm90/. Since the resampling procedure used an averaging filter, which is known to reduce high frequency noise, the relative elevation error should be smaller than that of the WorldDEM-3″.

### 2.2.8. WorldDEM$^{TM}$

The WorldDEM™ is the commercial digital terrain model (DTM) developed from the data acquired by the German TanDEM-X satellite program [17]. The WorldDEM™ is an edited version of the digital surface model (DSM) from which artefacts and objects located above the ground were removed [13]. The model's spatial resolution is 0.4″ (or approximately 12 m at the equator). The absolute vertical accuracy of the WorldDEM™ (LE 90% based on global product) is <10 m, but recent studies indicate that it is "outperforming the requirement by a factor of five" [6]. A separate study produced an estimate for the absolute vertical accuracy of the WorldDEM™ (the instrument- and environment-induced error sources only) at a level of <1 m [13].

### 2.2.9. Aeronautical Data on the Runway

In this project, the runway elevation data were also used to compare with the corresponding DEMs elevations. The aeronautical data on runways are in the public domain and available from the Aerodrome Obstacle Chart publication [12].



### 2.3. Data Processing

The RWYM was conceived in 2008 [16]. The RWYM assumes that the height error in a DEM is composed of three independent error sources: the instrument-induced error, the environment-induced error, and the target-induced error. This can be expressed as follows in Equation (1):

$$\sigma^2_{DEM} = \sigma^2_I + \sigma^2_E + \sigma^2_T \tag{1}$$

where $\sigma^2_*$ is the variance of DEM (*DEM*) error: instrument-induced (*I*), environment-induced (*E*), and target-induced (*T*).

While the instrument- (*I*) and environment-induced (*E*) errors can be managed by surveyors by selecting a more accurate instrument or mitigating the environmental conditions during data collections, the target-induced error (*T*) is due to the very nature of the terrain surface's discrete representation. As was demonstrated [16], the target-induced error can be estimated as follows in Equation (2):

$$\sigma^2_T = \frac{1}{12} d^2 tan^2(s) \tag{2}$$

where *d* is the DEM pixel size, and *s* is the terrain slope at a particular pixel.

The target-induced error is equal to zero when the slope is zero. This observation, along with Equation (1), leads to a directive that DEM vertical accuracy assessments should be performed on flat surfaces (slope = 0). Thus, only the instrument- and environment-induced errors will be captured, which is very important in the case of the SAR interferometry method of DEM production, simply because the whole process is extremely complicated. Hence, the instrument accuracy estimation is very difficult. In this investigation, 475 control points located on the runway's surface were used as the reference data. The elevation of the control points was obtained by line leveling. The bilinear interpolation method was used to calculate the corresponding elevations of the investigated DEMs. To visualize the discrepancies between the investigated DEMs and the reference elevations, centerline profiles of the differences were created, which allowed for a visual inspection and qualitative assessment of the discrepancies' behaviors.

The following data-processing steps were taken to estimate the vertical accuracy of the investigated DEMs (i.e., the instrument- and environment-induced error components):

1.  Using a bilinear interpolation method, the elevation for the locations corresponding to 475 control points were calculated for each investigated DEM;
2.  The discrepancies between the interpolated elevation and the control point elevation were calculated for each investigated DEM;
3.  The mean (*D*) and standard deviation ($\sigma$) of the differences were calculated for each DEM;
4.  The root mean square error (RMSE) of the differences for each DEM was calculated using the following formula:

$$RMSE = \sqrt{D^2 + \sigma^2} \tag{3}$$

5.  A histogram of differences was calculated for each DEM; and
6.  The Laplace probability density function (pdf) was calculated according to Equation (4):

$$f(x\,;\,m\,,\,a) = \frac{1}{2a} e^{-\frac{|x-m|}{a}} \tag{4}$$

The maximum likelihood estimator of *m* is the median of the differences, and the maximum likelihood estimator of *a* is given by Equation (5):

$$a = \frac{1}{n} \sum_{i=1}^{n} |x_i - m|, \tag{5}$$

where *n* is the number of samples (475 in this case).

## 3. Results

Table 3 shows the statistics of the differences between the investigated DEMs and the corresponding leveled spots on the runway. As expected, photogrammetric DEM is the most accurate. Its RMSE is approximately 6 cm, and its bias is equal to −0.002 m. The second-best DEM is the WorldDEM$^{TM}$, with a standard deviation of 0.327 m. However, this DEM exhibits a significant bias of −0.722 m. Hence, the RMSE is 0.787 m (or 1.291 m–LE90). Note that the TanDEM, in terms of its standard deviation of 0.231 m, is even more accurate than the WorldDEM$^{TM}$. However, it suffers from a larger bias of −0.923 m which, in terms of its RMSE, makes it slightly less accurate than the WorldDEM$^{TM}$. The third best-performing DEM is the AW3D30, with a standard deviation of 0.578 m, which is comparable to that of the WorldDEM$^{TM}$. However, the bias is 1.82 m, leading to an RMSE of 1.91 m (or 3.14 m–LE90). The worst-performing DEM is the SRTM-X, with a standard deviation of 5.209 m and a bias of −1.570 m.

**Table 3.** Statistics of the differences between the investigated DEMs and the leveled spots on the runway (DEM minus the reference elevation).

| DEM | Mean Diff. (Bias) (m) | STD (m) | RMSE (m) | LE90 (m) | Difference (m) | |
|---|---|---|---|---|---|---|
| | | | | | Maximum | Minimum |
| AP | -0.002 | 0.064 | 0.064 | 0.105 | 0.174 | -0.238 |
| ASTER | 2.435 | 1.992 | 3.146 | 5.159 | 7.583 | -3.341 |
| AW3D30 | 1.820 | 0.578 | 1.910 | 3.140 | 3.684 | 0.383 |
| SRTM-1″ | 0.580 | 0.990 | 1.147 | 1.882 | 3.392 | -1.652 |
| SRTM-3″ | 0.614 | 0.712 | 0.940 | 1.542 | 2.705 | -1.656 |
| SRTM-X | -1.570 | 5.209 | 5.440 | 8.922 | 13.631 | -14.977 |
| TanDEM | -0.923 | 0.231 | 0.951 | 1.560 | -0.233 | -1.286 |
| WorldDEM$^{TM}$ | -0.722 | 0.327 | 0.787 | 1.291 | 0.113 | -1.493 |

Abbreviations: STD: standard deviation; RMSE: root mean square error.

Figure 4 shows plots of the difference between each DEM and the leveling along the runway's centerline. Note the different vertical scale for the plots. Figure 4 allows for a visual verification that the DEMs behave "normally" (i.e., there is no apparent trend or outliers present in the investigated DEMs). This verification is necessary, because the traditional accuracy measures of the accuracy of DEMs (i.e., RSME, mean difference, and the standard deviation) do not allow for identification of spatially-dependent trend in the data.

Table 4 shows the parameters of the Laplace probability density function that were calculated using the maximum likelihood estimator [18]. These parameters were used to produce the Laplace pdf graph shown in Figure 5 on top of the histograms of the discrepancies between the investigated DEMs minus the reference elevations.

**Table 4.** Parameters of the Laplace probability distribution function estimated using the maximum likelihood estimator from the discrepancies between the investigated DEMs and the reference elevations.

| DEM | *m*–Median of Differences (m) | *a*–Equation (5) (m) |
|---|---|---|
| ASTER | 2.546 | 1.619 |
| AW3D30 | 1.774 | 0.439 |
| AP | -0.002 | 0.047 |
| SRTM-1″ | 0.568 | 0.813 |
| SRTM-3″ | 0.624 | 0.553 |
| SRTM-X | -1.875 | 3.694 |
| TanDEM | -1.494 | 0.200 |
| WorldDEM$^{TM}$ | -0.724 | 0.251 |

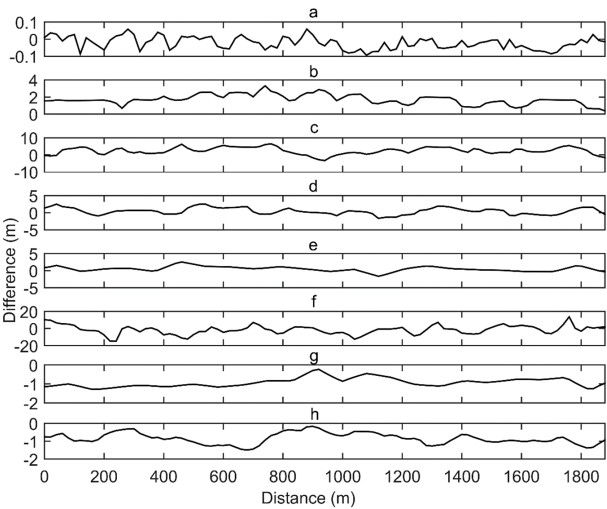

**Figure 4.** Plots of height differences between profiles extracted from the investigated DEMs minus the leveling along the runway's centerline: (**a**) aerial photogrammetry DEM; (**b**) ASTER; (**c**) AW3D30; (**d**) SRTM-1″; (**e**) SRTM-3″; (**f**) SRTM-X; (**g**) TanDEM; (**h**) WorldDEM^TM. Note the different vertical scales on the vertical axes.

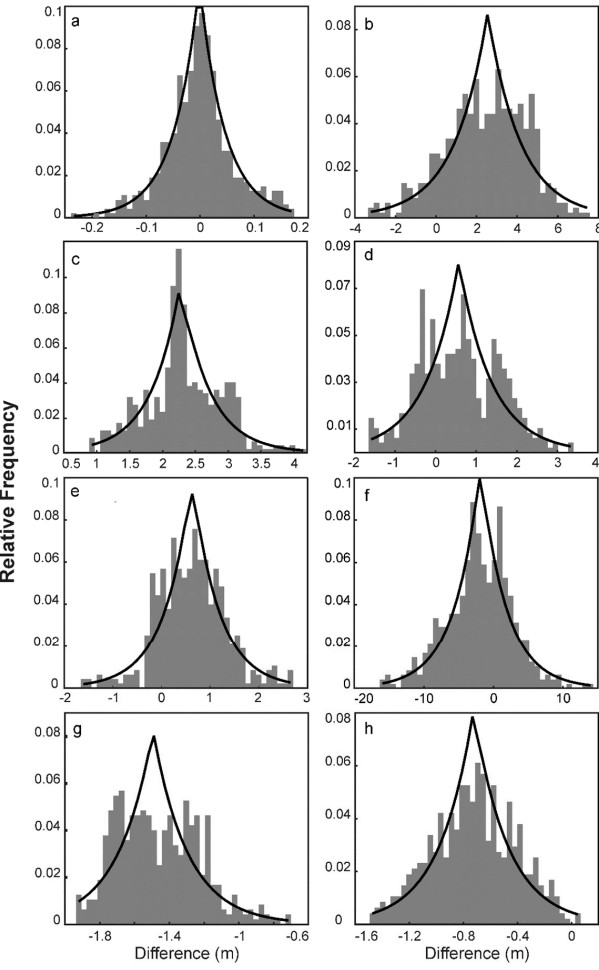

**Figure 5.** Histograms of the differences between the investigated DEMs and leveling: (**a**) aerial photogrammetry DEM; (**b**) ASTER; (**c**) AW3D30; (**d**) SRTM-1″; (**e**) SRTM-3″; (**f**) SRTM-X; (**g**) TanDEM; (**h**) WorldDEM^TM. A theoretical Laplace probability density function is also shown. The function parameters were estimated using the experimental data. They are shown in Table 4.

Figure 5 shows the histogram of the differences between the investigated DEMs and leveling; the Laplace probability density function is also shown.

## 4. Discussion

The comparison of the elevation of 475 control points evenly distributed across a runway's homogenous, almost flat surface provided an opportunity to assess the vertical accuracy of DEM models in a unique way. A typical approach to test a DEM employs sparsely distributed control points (e.g., GPS stations) located at various types of terrain cover and topography (slope). Alternatively, a higher accuracy DEM is used. However, the DEM vertical error model indicates that both terrain cover and topography also control DEM accuracy. Therefore, a flat and homogenous surface as a test bed largely eliminates the impact of these target- and environment-induced error sources. Hence, the results obtained in this study represent the DEMs' instrument-induced error sources, allowing a nonbiased comparison of the instruments' performance and the methods used to generate these DEMs.

Analyzing Table 3, one may conclude the following: (1) The bias in the case of photogrammetry-derived models (ASTER and AW3D30) is positive and reaches approximately 2 m. The bias for the X-Band InSAR-derived DEMs (SRTM-X, TanDEM, and WorldDEM$^{TM}$) is negative of the order of –1 m. Simultaneously, the elevation bias of the C-Band InSAR-derived DEM (SRTM-1"/3") is the lowest and of the order of 0.6 m. The global DEMs' bias issue was noted in the previous literature [2,19] and is believed to be due to the calibration of the SAR data (performed over the ocean), the accuracy of the ground control points (GCPs), and/or the accuracy of the geoid. This bias could be locally eliminated by estimating it using a few checkpoints with known elevations from an independent survey. (2) The standard deviation column in Table 3 represents the relative or point-to-point DEM accuracy and, therefore, is most important for engineering applications. The lowest level of the standard deviation is exhibited by the TanDEM and WorldDEM$^{TM}$, at 0.231 m and 0.327 m, respectively. Surprisingly, the lower resolution and better performance of TanDEM than that of the WorldDEM$^{TM}$ is a result of the averaging effect of the down-sampling procedure used to produce the 1" TanDEM model from the 0.4" original data. A similar effect can be observed in the case of the SRTM-3" vs. the SRTM-1" DEMs, for which the standard deviation is 0.990 m and 0.712 m, respectively. Judging by the numbers, one might conclude that a lower resolution DEM performs better than a higher resolution one (e.g., the SRTM-1" vs. the SRTM-3"). This deception can be explained by the fact that the results shown in Table 3 cover only the instrument-induced error source. In other words, they are good for a flat and horizontal surface. In the case of flat surfaces, the target-induced error component (Equation (2)) must be added. Thus, the slope of the terrain and the pixel size come into the play. Equation (2) clearly shows that, in the case of slope terrain, the accuracy is controlled by pixel size. A "break-even point" or a critical slope from which the higher resolution DEM is more accurate than the lower is approximately 1% or 0.613°. The third most accurate DEM in terms of standard deviation is the photogrammetry-derived AW3D30 model. Hence, this DEM's higher standard deviation is most likely caused by clouds—the major obstacle of the satellite-based photogrammetric method of DEM production. (3) In terms of the absolute vertical accuracy, the SRTM-3" and TanDEM exhibit almost equal sub-meter readings. However, because of the TanDEM's 1" resolution, its superiority over the SRTM-3" model will be evident in an even slightly undulated surface. The AW3D30 is significantly less accurate due to the level of bias, which can be easily locally estimated and compensated. (4) The graphs shown in Figure 4 do not reveal the existence of any clear trend, and they are also not correlated, suggesting that the differences are random and independent events. (5) The histograms of the differences shown in Figure 5 should resemble the Laplace probability density function, which can be visually verified. This is very visible in the case of the AP-derived DEM in particular. It should be noted that the results outlined in this paper are relevant for this particular man-made object.

## 5. Conclusions

The present study on the accuracy of the digital representation of the vertical dimension of a man-made structure using spaceborne DEMs can be concluded as follows:

1. It appears that the spaceborne InSAR technology is more accurate than the traditional photogrammetry based on the satellite imagery for DEM production. The limiting factor is the cloud cover that restricts the number of stereopairs used to develop DEMs (e.g., [2]).
2. The TanDEM dataset is slated to replace the SRTM model as a global DEM due to its higher vertical accuracy and the fact that it is more current than the almost 20-year-old SRTM dataset.
3. All the investigated DEMs, except the AP one, exhibited a vertical bias. The reason for the bias is probably related to the DEMs' vertical calibration. A separate study of the effect is being considered.
4. The vertical bias can be locally determined and subtracted from the DEM. This operation will increase the TanDEM's absolute accuracy to a level of approximately 0.5 m and the AW3D30 to 0.6 m (one sigma RMSE).
5. Both the TanDEM and AW3D30 spaceborne elevation data are good enough to perform at least preliminary studies on a variety of engineering projects.

The presented work contributes to the body of knowledge of surveying engineering and similar branches of technology and engineering disciplines, such that one of the most important quantitative representations of the topography of the Earth's surface, which is a DEM, offers a vertical accuracy level that can satisfy the needs of, at least, a preliminary study of a variety of projects, including hydrotechnical engineering, landscaping, and civil engineering. In addition, there is a clear trend in the availability of more accurate and high-spatial resolution DEMs, which will continue for the foreseeable future. Also, this work demonstrates the utility of the runway method of DEM accuracy assessment for the surveying community. This work also provides the first ever vertical accuracy assessment (on a local scale) of the latest DEM—the TanDEM.

**Supplementary Materials:** The following are available online at http://www.mdpi.com/2076-3263/9/9/387/s1, Dataset s1: DTM_Data.zip or [14]. The data file is in the csv format. The coordinates are referenced to WGS84/UTM36T.

**Author Contributions:** Conceptualization and methodology, K.B; validation, V.A., S.I., and Ç.M.; formal analysis, P.P.; data curation, P.P.; writing—original draft preparation, K.B.; writing—review and editing, P.P.; visualization, V.A. and S.I.; supervision and administration, Ç.M.

**Funding:** This research received no external funding.

**Acknowledgments:** The authors are grateful to Wolfgang Koppe of Airbus Defense and Space for providing the WorldDEM™ data over the AOI free of charge. The assistance of the Directorate General of Geographic Information Systems, Turkey for providing the aerial photogrammetry DTM is kindly acknowledged.

**Conflicts of Interest:** The authors declare no conflict of interest.

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
