# Peer review of "How Well can Spaceborne Digital Elevation Models Represent a Man-Made Structure: A Runway Case Study"

_geosciences, doi:10.3390/geosciences9090387_

Round 1
Reviewer 1 Report
This paper focuses on accuracy assessment of digital elevation models. Overall this work is of good quality (usefull) and fits with the scope of “Geosciences”. However, significant further work is required before the paper can be accepted for publication:
I suggest to add a Motivation (or Contribution) Explain in Motivation section, why have you conducted such research. Part of the text you will take from the Introduction section. I suggest you improve the statistical analysis: pp.7, line 211 “3. The mean (D) and standard deviation (σ) of the differences were calculated for each DEM” - I suggest you calculating the absolute value from the mean (average). This value shows the distribution better. Sometimes D is equal 0, otherwise |D| shows real average value of the discrepancies between the interpolated elevation and the control point elevation. “4. The root mean square error (RMSE) of the differences for each DEM was calculated using the following formula: RMSE= (D^2+ σ ^2)^0.5”.Where does this mathematical formula come from? (quotation) I have never seen such a mathematical formula. I recommend using the generally known formula notation RMSE= (1/N sum(f(xi,yi)-zi)^2)^0.5 where f(xi,yi)- elevation value of interpolation function at the point with coordinates x and y, z – elevation of control point.
In this case, RMSE and STD are similar. Don't forget to make any necessary changes to the text
“5. A histogram of differences was calculated for each DEM” . Please add Figure 5. The references should be improved, you can increase them.
Author Response
This paper focuses on accuracy assessment of digital elevation models. Overall this work is of good quality (useful) and fits with the scope of “Geosciences”. However, significant further work is required before the paper can be accepted for publication:
I thank you for your valuable time spent working on our manuscript. Below are our comments and answers to your specific questions and concerns.
I suggest to add a Motivation (or Contribution) Explain in Motivation section, why have you conducted such research. Part of the text you will take from the Introduction section.
Thank you for your suggestion, however, since we explicitly refer to the main research question already in the title of the paper: "How Well Can Spaceborne Digital Elevation Models Represent a Man-Made Structure: a Runway Case Study", we believe this clearly indicates the motive of our research, i.e., simply, we wish to answer the research question. In addition, in the Introduction we clearly state the aim (line 53-54) of the research, and the objectives (lines 68-73). We also point out that this research is an unique because it focuses on a single coherent man-made structure, and not on a regional or global assessment attempt of DEMs which typically is the case offered in a number of previous works.
-------------------------------
I suggest you improve the statistical analysis: pp.7, line 211 “3. The mean (D) and standard deviation (s) of the differences were calculated for each DEM” - I suggest you calculating the absolute value from the mean (average). This value shows the distribution better. Sometimes D is equal 0, otherwise |D| shows real average value of the discrepancies between the interpolated elevation and the control point elevation.
We use a standard statistical approach to describe the behaviour of a random process. This approach uses the first and second order moments, which are the mean value and standard deviation, respectively. This approach is exclusively used in this type of research, which allowed for comparison of results across various publications. In addition, both D and sigma (standard deviation - s) possess a clear and powerful physical interpretation which is: D = 'systhematic error or bias' of observations, and s = 'random error' of observations. Introducing another model of error for DEMs would be completely counterproductive in this case and in many other branches of geosciences.
-------------------------------
“4. The root mean square error (RMSE) of the differences for each DEM was calculated using the following formula: RMSE= (D^2+ s ^2)^0.5”. Where does this mathematical formula come from? (quotation) I have never seen such a mathematical formula. I recommend using the generally known formula notation RMSE= (1/N sum(f(xi,yi)-zi)^2)^0.5 where f(xi,yi)- elevation value of interpolation function at the point with coordinates x and y, z – elevation of control point.
In this case, RMSE and STD are similar.
The formula for RMSE we use in the manuscript has the advantage, that both D and s (sigma) possess a meaningful interpretation for geosciences professional. This is because D indicates the bias or systematic error present in the investigated dataset (e.g. DEM), and s is the standard deviation of the random error present in the dataset. The RMSE 'scrambles' both errors which is not very useful for in depth analysis of the accuracy of the investigated dataset. The formula used by us can be found in any elementary statistics textbook. Here it goes how we derived 'our' formula:
Let X be a random variable with average value E[X], where E[*] denotes the average operator. Then the standard deviation s can be calculated from the following formula: σ = (E[X2] - (E[X])2)0.5 (see any statistics textbook). Hence, as it can be easy shown:
E[X2] = (σ2 + (E[X])2)0.5.
where E[X2] is RMSE of X (in your case X is the difference f(xi,yi)-zi), and
(E[X])2 is square of the average value of discrepancies (in our case D2),
σ - is the standard deviation, which everybody knows how to calculate it.
Therefore, replacing E[X] by D and E[X2] by RMSE we finally get our expression for RMSE = (σ2 + D2)0.5
--------------------
Don't forget to make any necessary changes to the text
Thank you for your kind reminder. We certainly will.
--------------------
“5. A histogram of differences was calculated for each DEM” . Please add Figure 5.
We do not understand your concern, because Fig. 5 is included in the manuscript.
--------------------
The references should be improved, you can increase them.
A current number of publications on spaceborne DEMs is approx. 5000 items. We had to make a compromise to safe space vs. expanded reference list. Therefore, any subset of items from such a large body of publications would have subjective character as this is the case in with manuscript. Thank you for understanding.
Reviewer 2 Report
Review of the paper “How well can spaceborne digital elevation models represent a man-made structure: a runway case study” by Becek et al.
The manuscript discusses assessment of the remotely derived digital elevation models over a small study area located in an airport line in Turkey. The manuscript is well written and enjoyable to read. Despite the fact that topic is not innovative, it is worth investigating, specially because nowadays, many of the surveying project tend to use global DEMs rather local surveying due its low (free) cost. The outcome of the manuscript indicates that the global models are only good for primarily phases of the civil projects and in fact, a precise leveling need to be done for the next steps. The only major comment that I have is the study area is too small for assessing the globalDEMs. In fact, authors’ conclusion can be misinterpreted that their results are good for everywhere in the world, whereas this is not true. Comparing the global DEMs with only 475 control points mostly located on a profile (only 30m width in the runway) will not give a reliable result for overall assessment of DEMs. I understand the concept behind the manuscript, and I hope authors did not intend to compare the DEMs and they only wanted to assess them against their study points (as the title of the manuscript says so). However, there are a few places in the manuscript which authors do not make the point clear that their assessment is only good for their study area and it is not an overall comparison of DEMs. Some of these places are:
Abstract: “The most accurate DEM …”: this is in fact true only for the test area, I recommend adding the “for our test area” to the beginning of this sentence to prevent any misunderstanding to readers. The motivation part of the introduction (P2 paragraph 2) where authors compare their work in the present study with other DEM assessment articles. In the conclusion.
If authors tend to show the assessing results of available DEMs against their man-made structure not DEMs themselves, then they should make this clearer in the text. If the case is the latter point, then I believe such a small control points cannot be used to judge DEMs against each other.
There is one more part missing in the paper which is about the DEMs used in the study. I recommend authors adding a table showing the details of reference frames (horizontal-vertical) of the DEMs they used and also write whether their national benchmark they used for their precise levelling has any bias with respect to the vertical reference frames used in the DEMs. That could help explaining the existing bias in their comparison.
Overall, I think the paper can be published after applying aforementioned comments.
Author Response
The manuscript discusses assessment of the remotely derived digital elevation models over a small study area located in an airport line in Turkey. The manuscript is well written and enjoyable to read. Despite the fact that topic is not innovative, it is worth investigating, specially because nowadays, many of the surveying project tend to use global DEMs rather local surveying due its low (free) cost. The outcome of the manuscript indicates that the global models are only good for primarily phases of the civil projects and in fact, a precise leveling need to be done for the next steps. The only major comment that I have is the study area is too small for assessing the globalDEMs. In fact, authors’ conclusion can be misinterpreted that their results are good for everywhere in the world, whereas this is not true. Comparing the global DEMs with only 475 control points mostly located on a profile (only 30m width in the runway) will not give a reliable result for overall assessment of DEMs. I understand the concept behind the manuscript, and I hope authors did not intend to compare the DEMs and they only wanted to assess them against their study points (as the title of the manuscript says so). However, there are a few places in the manuscript which authors do not make the point clear that their assessment is only good for their study area and it is not an overall comparison of DEMs. Some of these places are:
Abstract: “The most accurate DEM …”: this is in fact true only for the test area, I recommend adding the “for our test area” to the beginning of this sentence to prevent any misunderstanding to readers. The motivation part of the introduction (P2 paragraph 2) where authors compare their work in the present study with other DEM assessment articles. In the conclusion.
If authors tend to show the assessing results of available DEMs against their man-made structure not DEMs themselves, then they should make this clearer in the text. If the case is the latter point, then I believe such a small control points cannot be used to judge DEMs against each other.
------------------
Thank you for your careful assessment of our contribution. You concern that some of the readers maybe under impression that this study is about global vertical accuracy assessment of DEMs is most likely overstated, because we explicitly state in the title, that this is a case study i.e., valid for the particular runway only. But, please note, (we didn't say that in the paper) our results are very similar to results of other authors obtained during regional or global assessments of DEMs in question. This clearly indicates that the DEMs are of consistent and non-spatially dependant quality.
As suggested by you, we have altered the text to address your concerns.
-------------------
There is one more part missing in the paper which is about the DEMs used in the study. I recommend authors adding a table showing the details of reference frames (horizontal-vertical) of the DEMs they used and also write whether their national benchmark they used for their precise levelling has any bias with respect to the vertical reference frames used in the DEMs. That could help explaining the existing bias in their comparison.
Table 2 is now included as requested.